# Second Partnership and Dementia Care in a Blended Family: Case Study of a Wicked Problem

**DOI:** 10.3390/ijerph21091213

**Published:** 2024-09-16

**Authors:** Olga Asrun Stefansdottir, Merrie J. Kaas, Tobba Sudmann

**Affiliations:** 1School of Health, Business and Natural Sciences, University of Akureyri, Solborg v/Nordurslod 2, 600 Akureyri, Iceland; olgastef@unak.is; 2School of Nursing, University of Minnesota, 308 Harvard Street SE, Minneapolis, MN 55455, USA; kaasx002@umn.edu; 3Department of Health and Social Sciences, Western Norway University of Applied Sciences, Campus Bergen, N-5020 Bergen, Norway

**Keywords:** second partnership, blended family, dementia care, family caregiving, case study, wicked problem

## Abstract

Dementia care research has largely ignored the challenges that may emerge from couple and family dynamics, especially about second partnerships in blended families. This paper details the case of a 79-year-old man, Hannes, in his second partnership who tried to handle the complexities of his wife’s dementia care as her children and healthcare providers discounted his role as husband and decision maker. He faced difficult communications with family members and challenges to his role as decision maker by healthcare providers and legal system professionals. This man’s story is explored through multiple interviews and document analyses from 2020 to 2023. This case study uses the concept of the “wicked problem” to frame the potential complexities of dementia care when blended families are involved in decision making. This framework allows us to consider the many facets of family dementia care and how improvements could be made to facilitate role transitions and family decision making.

## 1. Introduction

A less understood issue in professional dementia care and support is understanding and responding to families who have disagreements about the care of a parent with dementia. Dementia care research has largely ignored the challenges that may emerge from couple and family dynamics, especially in blended families, when care decisions and economic decisions need to be made. Conflicting perspectives among family members about the care needs of a person with dementia can cause wide-ranging effects on marriage, family life, family-based care and burdens, and confusing communication with formal care providers in a home or institution [1,2,3].

### 1.1. The Grip of the Couple Norm

Couple norms have an insidious grip on our lives, as they are at the heart of how intimate life is organized, regulated, and recognized in Western countries [4]. Additionally, Roseneil and co-authors say, “Whether a person is coupled or not is fundamental to their experience of social recognition and belonging” [4] (p. 4). This may be true for people in their fertile years, but there is no less of a need to consider the need for intimacy and belonging to others when people are in old age. Following Daniels and co-authors, it is essential not to forget that behind failing physical and cognitive health, there is still a living partner relationship. Current partners may also have children from previous marriages [5]. Therefore, as functional decline or dementia arises more often in later partnerships, disagreements about who provides support might increase compared with those have been in one partnership [2,6,7,8]. This case study is cognate to our earlier research, where we have observed that the flip side of patient-centered care is that the present partner’s needs are ignored [3,9,10], and even more so with older adults who have had previous marriages [2,3].

### 1.2. Navigating Blended Families and Second Partnerships

When people remarry, or partner, and one partner has a child from an earlier relationship, it is called a stepfamily [11]. If both partners have children from previous partnerships, it is called a complex stepfamily [12]. In late-life complex step or blended families, children typically no longer live with their parents or may have distant relationships with them [13]. This is the case in Hannes’s and Anna’s marriage. For consistency’s sake, we will use the term blended family, which is appropriate as none of their adult children have ever lived with them. At face value, contemporary family formation patterns differ from earlier generations when couples grew old with their first and only spouse, or they remarried after the death of a spouse. Today, many forms of cohabitation exist with and without the legalities of marriage. Remarriage and the formation of new families create challenges. New and different families can require other arrangements of decision making for the older couple, e.g., about inheritance or caregiving in later life, facing diminishing health and function, like dementia [14]. Older couples with children from a previous partnership may also have to face challenges as to the question of whose children are willing to provide support to the person in need or if they only support their own parent [15,16].

### 1.3. Navigating Dementia Challenges within Blended Families

Dementia is more than an individual disease and experience, it is also a family illness affecting the family dynamics and represents a shifting of the responsibilities and challenges of everyday life from both partners to the “healthy” partner [1,2,10]. As with any significant illness, the need for communication among increasing numbers of people, places, and systems can be taxing, not only for the persons with dementia but also for partners and family [17]. Culturally, what is recognized as a family varies by time and place [13,14]. Often, bloodlines have more influence, whereas others in the family have less. Country-specific rules and statutes for marriage, inheritance and ownership vary. Family rights to make health care decisions may be at odds with the legal or health care system when professional care is needed. In the case of blended families, questions can arise about who is responsible for care and decision making [1]. Understanding blended family functioning, particularly relationships between older parents and their adult stepchildren, is vital to appraising the viability of stepfamilies in providing care to older adults [13].

### 1.4. “The Hidden Struggles of Dementia Caregivers: A Wicked Problem?”

Previous research has shown that family caregivers of people with dementia experience a decrease in life satisfaction due to emotional, social, and economic reasons [18], as well as inadequate support from formal medical and care services [19]. Family caregivers also express challenges such as grieving for the loss of a relative’s personality, exhaustion from caregiving [20], feelings of being forgotten and alone in their struggle to care for their relatives and of being unheard when speaking up for their loved ones when their needs are ignored [9,21]. In Iceland, where this case study took place, the emphasis on services for people with dementia and their families is on diagnosing and evaluating the disease’s progress and treatment. The tasks of daily life are viewed as the family’s responsibility [7]. Using the concept of a wicked problem, the case presented here is used as an example to frame the challenges of dementia care when blended families are involved in decision making [22]. This framework allows us to consider the many facets of family dementia care and how improvements in healthcare services could be made to facilitate role transitions (from husband to carer and from wife to service recipient) and family decision making. Wicked problems, defined by Rittel and Webber [22], are complex social or cultural problems with an unknown number of potential solutions. This can be an interconnected issue that defies straightforward solutions due to its multifaceted nature and the involvement of various stakeholders. In this article, we consider how wicked this case study is using their definition (Table 1).

This article illustrates dementia caregiving from the experiences and perspectives of Hannes, a 79-year-old man in his second marriage. His life started to fall apart when his wife developed dementia, and her children disagreed with him about who was the rightful decision maker and how to organize her care and on her behalf. In agreement with the narrator (Hannes), we aim to share his story as a paradigmatic case, illustrating how covert or muted conflicts and opinions among extended family members arise. The following research questions guided the analysis of the case: Which challenges does an older husband in his second marriage meet considering his wife’s increasing cognitive decline? Who is involved in his wife’s everyday care, and how do they relate to his needs and wants?

This family story can be seen as a wicked problem in dementia care because the stakeholders and family members disagree about the problems they face and possible solutions, and who should make the final decisions about family caregiving solutions.

## 2. Materials and Methods

### 2.1. Setting, Participants, and Recruitment

This study emerged from more extensive qualitative comparative research on ageing, dementia and partnership in Norway and Iceland (2017–2020). Icelandic participants were invited through staff leaders (gatekeepers) in long-term care facility (LTCFs), and the Norwegian participants were invited to participate through advertisements in a local newspaper. From the comparative study (*n =* 25), eligible participants who agreed to participate were over 67 years old, with experience caring for their spouse with dementia at home and had a spouse at LTCF. The interviews (face to face, one with each participant) were conducted at the participant’s home and lasted 1–2 h. Interviews were digitally recorded and transcribed verbatim. Software NVivo 11 [23] was used to manage raw data.

The present case is an in-depth analysis of one paradigmatic case produced from the more extensive comparative study. During the data production, we realized 4 out of 25 participants faced challenges that were not present in the remaining empirical material. They told stories from their second marriage and belonged to a blended family. When their spouse’s care needs increased, these partners experienced difficulties communicating and collaborating with family members and of having their views recognized by healthcare professionals. These four partners from second partnerships seemed to lose inter-personal connection as a spouse with their last partner more rapidly than did partners in a single partnership. The differences between single marriage partnerships and multi-marriage partnerships were striking. The lack of research focusing on older age and multi-relationships prompted us to undertake this case study.

A narrative case study was used to conduct detailed research, understand the disease progression stages, and investigate the couple’s situation within its environmental context [24]. Such qualitative case studies can illustrate a unique, unusually interesting case that needs to be described and detailed [25]. It can capture essential features and meanings that might not be as effectively captured by other research designs and can provide access to information that might otherwise be difficult to study [26].

This article presents a case story from Hannes (pseudonym), who lives in Iceland with his second partner Anna (pseudonym), who was diagnosed with Alzheimer’s and subsequently admitted to a nursing home. Alzheimer’s disease is a progressive neurodegenerative disease of the brain which cannot be cured, and where the inevitable outcome is death. The disease progression is varied, but changes to physical, mental, social and cognitive functions are constantly progressing [27]. The first author of this paper met and talked to Hannes several times over the period 2020–2023, including three audio-recorded face-to-face interviews, phone calls and emails, access to his written diary, and official letters and documents of relevance to his marriage and his wife’s transition to the nursing home. The empirical material gave us access to rich and in-depth data to be analyzed as a paradigmatic single case [25]. The first author conducted all of the conversations and interviews with Hannes in Icelandic.

What follows will first be a short presentation of Hannes’s and Anna’s story, intended to be the necessary background material for the analysis and discussion.

### 2.2. Hannes’s and Anna’s Story—A Brief Introduction

Hannes started to write a diary about Anna’s Alzheimer’s story in 2020. It was his way of keeping track of their partnership and all communications concerning caring for Anna and her move from home to nursing home. The couple became friends for four years before they started to live together. They both had been married and divorced previously. Hannes had three children from his previous marriage, while Anna had four. After two years of living together, they married in February 1998 when Hannes was 55 years old and Anna was 57 years old. They both owned properties, but when they got married they lived in Hannes’s house for two years, and two of Anna’s children lived in her house without paying rent. Then, the couple decided to sell Hannes’s property and move to Anna’s house. At that point, Anna’s adult children then needed to move elsewhere. How happy Anna’s children were to move elsewhere was not reflected in Hannes’s story. Hannes and Anna chose to pay Anna’s house debts from the sale of Hannes’s house and put the rest of the money in a bank to fund their retirement.

Hannes characterized their marriage as happy and described what great friends they had always been. According to Hannes, occasionally throughout their marriage, he and Anna held family parties during which his children and Anna’s children and grandchildren met and ate together. However, more often they met with each other’s children separately. Hannes and Anna had a summer house that Anna bought before they married, and her children, especially one of her daughters and her family, used it often. Hannes and Anna often went there throughout their marriage and they also travelled abroad. In 2015 Anna was diagnosed with Alzheimer’s. In the following years, her health quickly deteriorated, and Hannes noticed changes in his wife’s behaviors, like remembering and repeating the same story many times.

Hannes’s children and two of Anna’s children sometimes travelled abroad with them together or individually to assist them with the journey. Hannes paid for their fare and accommodation. However, at some point, Hannes realized that Anna’s daughters were subverting his and Anna’s wants and needs instead of assisting him and their mother to carry out their travel plans, so he stopped asking them for help. During 2017–2019, Anna’s dementia and care needs increased considerably. Eventually, healthcare professionals, friends, and neighbors contributed to Anna’s care. During these years, Hannes realized he had to seek legal assistance regarding his ownership of his and Anna’s property and inheritance concerns for his children due to emerging conflicts with her children.

During the interviews, Hannes expressed gratitude for their 20 excellent married years together but also said it was difficult to watch his partner go into “oblivion,” as he calls it, and he felt great sadness at having to admit her to a nursing home.

### 2.3. Ethical Consideration

When inviting people in vulnerable situations to take part in research, like older partners taking care of spouses with dementia, it is mandatory to secure the participant’s informed consent to participation and ensure that they trust our confidentiality. Furthermore, participants must be informed about how the research may impact them and which roles the researchers may have [28]. Therefore, formal ethical permissions to undertake this study from the Norwegian Centre for Research Data (NSD-48366) and the Icelandic National Bioethics Committee (VSN-18–149) were obtained in parallel in 2017–2020. The Icelandic participants provided written informed consent, and the Norwegian participants gave oral consent, as stated at the beginning of the interview recording. All participants were informed about the research aims and assured that their identities and records would remain confidential, and they were free to leave the study at any stage. An agreement was made with a specialist in psychiatric nursing to give the participants one hour of counselling free of charge if they felt they needed it after participating in the study.

### 2.4. Analytical Strategies

Co-authors brought different experiences and qualifications to the study: the first author is an occupational and family therapist, the second is a psychiatric nurse with special certification in gerontology, and the last is a physiotherapist and medical sociologist. All are qualified qualitative female researchers on ageing and cognitive decline. In conjunction with the concept of a wicked problem, data analysis for this article was performed through the theoretical lenses of critical gerontology and family systems theory. Critical gerontology provides researchers with tools by which to critically appraise and uncover more detail of cases or situations. For example, in this case study, we discovered the impact of family conflict when caring for a spouse with progressive cognitive decline. This adds a critical sociological approach to the analysis of the material, cultural, and contextual conditions and circumstances that affect partnerships in later life [29,30]. We applied a theoretical lens from family systems theory to the couple’s stories and the analytical documents [31] to provide insight into the family background story, their relationships and their interactions.

The first step in the analytical process was to transcribe interviews and organize the transcripts and the case-relevant documents into a timeline (Figure 1). The figure describes when Anna is diagnosed with dementia and the challenges Hannes faces in caring for her, and the number and types of people who were involved in that process. The timeline was analyzed to identify significant moments where fallouts, disagreements, muted issues, or incompatible wants and needs emerged.

## 3. Findings

The three interviews and the various document analyses were split into the following five different periods of Hannes’s experience of being a second marriage partner when his wife developed dementia: (a) “This has been a great stress for me”, i.e., how many people are walking around Hannes’s and Anna’s intimate space; (b) “I was not exactly welcomed”, i.e., being in between acceptance of the partnership and rejection within the family; (c) “Things were out of my hands”, i.e., who makes decisions regarding Anna’s care; (d) “I am ordered and threatened”, i.e., the experience of being an insider in your marriage to becoming an outcast; and (e) “I cannot do anything”, i.e., navigating the challenges of a loved one’s transition to a nursing home.

### 3.1. “This Has Been a Great Stress for Me”: How Many People Are Walking around in Hannes’s and Anna’s Intimate Space?

Hannes described how Anna’s health deteriorated between 2015–2017, and in 2017, she was offered dementia daycare services, which she refused. In 2018, Hannes contacted the healthcare system and the family for more help caring for her because at that time he had been diagnosed with cancer for which he needed treatment. In the continuing months, Anna was again offered a two-week stay in a dementia daycare center. She did not want to go there, no matter how much Hannes or her geriatrician tried to convince her that it would benefit her. When Hannes went to his cancer treatment, he needed to leave Anna alone in their house and was very apprehensive about her safety. Hannes finally received some help from the health care system 2–3 times a week when a home care nurse came to be with Anna while he received his cancer treatments.

Anna’s and Hannes’s children significantly impacted her care, especially between 2018 and 2020. Hannes’s children, especially his daughter, supported him in caring for Anna with practical things like swimming trips. Anna’s children also supported him in her care by going with them during trips abroad and sometimes staying with them at their home. At times, Anna’s daughters came to Hannes’s and Anna’s house to calm her down when she did not recognize Hannes and wanted to throw him out of the house. This happened repeatedly between 2019 and 2020. Even though Anna’s children were willing to help Hannes with their mother’s care, they disagreed with him about their mother’s attendance at dementia daycare. They said their mother should not be in this daycare because she would do much better at home. The couple’s friends, especially their swimming partners and neighbors, were also involved in Anna’s care, including practical things like dressing and bathing her, looking after her when Hannes was gone, and providing emotional support to Hannes when needed. Many healthcare professionals also engaged with Hannes in Anna’s care from 2018 to 2020. All of these new, additional people were significant changes for Hannes, who had mainly been alone as a caregiver before this time. In a way, these interactions confused Hannes and caused him more stress because of everyone’s different ideas about the proper care for Anna.

In October 2020, their swimming partners became aware of the fact that Anna thought Hannes was not her husband. More events like this happened every day and people noticed them. Anna sometimes claimed that Hannes was her ex-husband, who had physically and mentally abused her throughout their 25-year marriage. Many nights she asked where Hannes lived and eventually kicked him out of the house. Hannes’s response was to show her wedding photos and old photo galleries with pictures of their marriage. When nothing worked to calm her, it always ended the same: he had to leave the house. The only thing Hannes could do in that situation was to call one of his stepchildren to ask them to come and calm their mother down. When they came, he left the house for half an hour and when he came back, Anna did not remember why he had left and was always happy to see him.

In November 2020, the nursing home administration ruled that Anna was deemed suitable for respite care or even a permanent place in a nursing home. By then, Hannes realized how bad her cognitive deficits were and in consultation with the head of the respite care home, he decided to accept respite care for his wife. Her increasing needs were starting to drain him mentally and physically. Hannes was also told by the nursing home administration that Anna could get a permanent place in another nursing home directly following the respite care due to the advanced stage of her Alzheimer’s, which he decided to accept. Hannes said that he somehow realized that he was like a prisoner in his own home caring for his wife and that his health was also increasingly deteriorating.

### 3.2. “I Was Not Exactly Welcomed”: Being in between Acceptance of the Partnership and Rejection within the Family

According to Hannes, Anna had been in a very violent marriage with her first husband. As he explained, her children were also abused by their father and when Anna divorced him, the children moved with their mother. Hannes described the relationship between him and Anna’s children as very good when they became a couple. However, when he married Anna, he sensed that her children felt he was taking her away from them.


*“I felt guilty that I was taking their mother away from them (when he married Anna). That was my feeling in the beginning. I was not exactly welcomed, but we all adapted to each other, and things turned normal as I felt.”*
(Hannes, interview II)

Initially, Hannes felt unwelcome by Anna’s children, but gradually his relationship with them developed more positively. Hannes described occasional family gatherings throughout their marriage years where the blended family (his children and Anna’s children) gathered even though their children were never particularly close to each other. Nevertheless, suddenly everything changed in Hanne’s and Anna’s children’s communication the same day Anna entered respite care services.


*“I want to mention that my interactions with Anna’s children had always been ordinary until I admitted her to a nursing home for rest without consulting the children since I had every right to that. It was done in consultation with healthcare professionals.”*
(Hannes, interview II)

In discussion with the healthcare professionals, Hannes asked if he could accept this respite care for his wife without asking her children their opinion. The healthcare staff said, *“Why do you ask? You are her husband.”* Hannes then decided to make this decision alone because he was so exhausted from caring for his wife that he needed a rest. Anna’s children were never fond of respite care for their mother. Hannes described the sequence of events as follows: as soon as he had driven Anna to the respite care home, he immediately informed her children. They reacted angrily to Hannes for not consulting them before admitting their mother to the respite care home because they did not believe it was time for their mother to be admitted based on her cognitive status. Even though Hannes told Anna’s children that he had made the decision after consulting with the health professionals, they did not believe him. The day after Anna went into respite care two of her children visited her. After the visit, they told Hannes that their mother was unhappy, the place was miserable, and their mother wanted to go home. Three days later Anna’s children decided to take their mother from the respite care home, even though Hannes and the primary nurse disagreed and tried to tell them that both Anna and Hannes needed this respite care for two weeks. However, neither Hannes nor the healthcare staff could prevent Anna’s children from removing her from the respite care home.

After this happened, the relationship between Hannes and Anna’s children deteriorated. Immediately after Anna’s children took her out of respite care, they wanted her to go home to Hannes. Hannes refused and said the plan was for Anna to get this respite care for the next two weeks so he could also rest from his cancer treatments. Consequently, Anna stayed at two of her children’s homes alternately, with Hannes allowed only limited contact with his wife. After two weeks, Anna’s children returned her to Hannes, and Hannes saw that Anna’s condition had deteriorated even more.

In mid-December 2020, Hannes wanted Anna to go immediately to the permanent nursing home where her move had been approved. Again, Anna’s children were not ready to accept Anna’s worsening dementia and reconsider their mother’s need for nursing home care. During this time, COVID-19 was rampant, and visitation restrictions were imposed. The primary nurse of the nursing home (where Anna was accepted for a permanent stay) asked Hannes to wait until the end of the year to mediate between him and Anna’s children about her admission. Hannes agreed with the request and kept Anna at home. Christmas was mainly quiet for them. Anna’s children invited their mother to dinner at Christmas, but Hannes was not invited. They told their mother: *“Hannes is not welcome in this family.” (Hannes, interview I)*.

### 3.3. “Things Were Out of My Hands”: Who Makes Decisions about Anna’s Care?

At the end of December 2020, Hannes took Anna to the new nursing home for permanent admission. Again, her children disagreed with him about Anna’s admission, so they picked their mother up the next day and took her to one of their homes. With that, things were again out of Hannes’s control. Anna did not move to the permanent nursing home until the end of January 2021, which was then her children’s decision. However, during her three-week stay at her children’s home in January 2021, leading up to her move to nursing home, Hannes received a letter from the Sheriff stating that Anna wanted to divorce him. Hannes was very startled and confused and did not understand what was happening. In the interview, he described it this way:


*“By then, Anna’s children had arranged an interview with the Sheriff and told me that Anna and I should divorce, (Hannes was speechless). He answered Anna’s children by asking: Shall we divorce? There is nothing that can separate us. Then I discovered that Anna’s children had booked an interview with the Sheriff, which had happened three weeks before. The Sheriff told me that later. I received an email asking me to attend a meeting with the magistrate regarding Anna’s divorce petition on the same day the meeting was supposed to occur. So, I did not have a chance to attend. It was then entered in the meeting book: Hannes had not arrived nor announced his absence. However, I knew nothing about this.”*
(Hannes, interview I)

Also in January 2021, a letter with a new electronic identity card for Anna arrived at their home. Hannes was shocked because he knew that Anna’s doctor had confirmed that she could neither file for divorce nor apply for new electronic ID cards. Hannes also realized that, with an electronic identity card for Anna, it would be possible for others besides himself and Anna to access all their joint funds. Hannes immediately informed the company that provides such certificates that his wife could never have done this alone, and therefore it could be a matter of breaking the law. He requested that these electronic IDs be destroyed immediately, and the company accepted his request.

### 3.4. ”I Am Ordered and Threatened”: The Experience of Being an Insider in Your Marriage to Becoming an Outcast

During 2021–2022, Hannes experienced constant challenges. He and Anna had shared their finances since marriage, and he managed them. When he realized that more than the two of them could enter their private financial bank accounts because of Anna’s new electric ID, he wrote a letter to the district court with the help of his lawyer to stop Anna from accessing their finances. Hannes had confirmation from Anna’s geriatrician that she was no longer able to make cognitive decisions due to advanced Alzheimer’s and therefore could not manage her finances either. At this time Hannes also transferred funds from Anna’s account to his to protect their assets.


*“I had my wife’s finances suspended because her children were trying to get control over our assets and were, among other things, getting her bank balances in their hands. I was protecting our property.”*
(Hannes, interview I)

Because of a letter Hannes received from the magistrate, he knew that Anna’s children feared he would not provide their mother with the necessary financial resources to provide personal care, and they feared he would also unreasonably diminish her interest in their joint property to the detriment of her estate. Anna’s children stated this in a letter to the magistrate requesting their mother obtain a legal guardian who would not be Hannes. It was one more shock for Hannes when Anna was assigned to a new legal guardian. Subsequently, Anna’s legal guardian requested that Hannes sever his financial partnership with Anna and demanded a key to Hannes’s home so that, ostensibly, Anna could go there whenever she wanted or at least once a week. Hannes’s response was:


*“I complained to the senior guardian and wanted the current guardian removed, and I would take over the job, but my request was rejected and dismissed on the grounds that I was not a party to the case. I am a party to the case because my interests and assets are being manipulated. I am ordered and threatened.”*
(Hannes, interview II)

After this first letter, Hannes later received another letter from Anna’s legal guardian, signed by his wife, in which he was accused of embezzlement. However, Hannes said he had no idea how Anna’s legal guardian had gained access to Anna’s finances. In addition, Anna’s children, who asked the legal guardian to obtain a house key on behalf of their mother, said it was their mothers right to have keys to their home. Hannes refused. He told Anna’s children that their mother could always come home when he was home. He knew that Anna’s children were only asking for keys for them to enter his home, not their mother.

Hannes described the chaos and difficulty of handling all those things. He was sad and disappointed because he and Anna had always been in love and had a close relationship. Hannes decided to change the lock on the front door of his home to ensure that no one could get in when he was not home. Then Anna’s children asked the police to talk to Hannes to get obtain key to his house. However, when the police came to Hannes’s house, he told them he did not want people entering his house if he was not home.


*“I have never prevented Anna or her children from coming to our house. The police should be able to testify to that. In January 2021, Anna’s children sent the police to me to demand new keys. When I told the police that Anna was welcome, but I did not want others to enter my home in my absence, the case was dropped. There are values and objects, e.g., computers, which I do not allow strangers to access.”*
(Hannes, interview I)

After Hannes explained his situation to the police, the case was dropped, but he faced additional challenges interacting with Anna’s legal guardian. There were various documents regarding access rights to the summer home. Anna’s legal guardian asked Hannes to give up ownership of the summer home to Anna and her children, but he refused. Hannes planned to stay in the summer house but would allow Anna’s children to go to the summer house if they asked him for permission. An example of one of these interactions was a letter to Hannes from Anna’s legal guardian with the following text:


*“Firstly, the undersigned (Anna’s trustee) requests that, on behalf of Anna, she and her children be granted access to her personal belongings and clothing.*



*Secondly, Anna must be allowed to come to her house once a week, whether you stay at home or not. It is suggested that there be a specific day a week when her children will bring her and stay for 2–3 h every time. At the same time, Anna will have the use of her holiday home every other week from Wednesday to Wednesday, whether her children will stay there, or she will stay there by herself”.*



*Thirdly, you must accept paying real estate taxes and other expenses on your shared property, as Anna no longer resides there.*



*Fourthly, you are required to agree to the liquidation of the financial partnership between you and Anna and that you shall not request an official exchange”.*
(A statement from a letter to Hannes from Anna’s legal guardian)

### 3.5. “I Cannot Do Anything”: Navigating the Challenges of a Loved One’s Transition to a Nursing Home

Hannes reflected on the emotional journey of the past three years since Anna moved to the nursing home in 2020. Amidst the changes, he described how he grappled with what he said was the most challenging: finding a new rhythm in life while living apart from his wife. By seeking support from various avenues, including priests and psychologists, he gradually coped with the trauma of losing their shared home space. All of this, he believes, has benefited him greatly by improving his health, and primarily helping him recover from all the traumas related to the invasion and destruction of his private space. When he was asked about his interaction with the healthcare system regarding Anna’s care, he answered:


*“I am glad you mentioned the system and communication. Although Anna’s children are like this to us, I cannot do anything. I am advised not to answer and will not argue with them. There is a need for an official body that takes a case like ours on their shoulder. I had one meeting with the doctor, which I requested when my wife entered the nursing home, but nothing came of it; I felt he was defensive. I have never been called to meetings (regarding Anna) for any reason. They are probably just doing their job. I need to be taken care of and the issues discussed frankly, but not in such a way that I am considered a rambler.”*
(Hannes, interview III)

Despite feeling guilty about not being able to do more to care for Anna, Hannes emphasized the need for better support for spouses and families dealing with dementia. Family information is missing regarding the progress of dementia, what steps to take, what services are available and the best time to use the available services. Hannes wanted and needed a coordinated plan for guiding the families of people with dementia through the challenges of dementia care. Hannes would have preferred to receive more spiritual help after Anna moved into the nursing home to help him adjust and accept the changes in their relationship. Hannes visits Anna three times a week and these visits bring him moments of joy, even if Anna does not express missing him explicitly.


*“I do not feel she misses me, but she is always pleased to see me. It is wonderful, and now she is starting to feel good. It helps to see that she is happy and content with the nursing home, which makes me feel good, too.”*
(Hannes, interview III)

## 4. Discussion

In the case of Hannes and his family, we explored the challenges he faced as an older husband in his second marriage as his wife, Anna, experienced increasing cognitive and physical decline. Their story raises many questions about dementia care and family system dynamics. We consider whether their situation qualifies as a “wicked problem” and whether through this lens of “wicked problems,” we can effectively capture the challenges faced by Hannes and his family members. We argue that, by recounting this family’s emotional, social, and practical caregiving needs we can shine a light on the need for better support and understanding within current healthcare services. See Table A1 in Appendix A for a more complete description of this case as a wicked problem.

Wicked problems are defined as “wicked” because there are no easy or right solutions. Wicked problems are complex and unique, yet in many cases, they are symptomatic of underlying problems or past issues [22]. In Hannes’s case, his and Anne’s blended family did not define themselves as a “family.” They were adults brought together by their parents’ marriage. Maybe this is not unusual in later life re-marriages or partnerships. Anna’s children continued to view her as their responsibility as their mother’s dementia worsened, as did Hannes. Their case describes trust and communication issues between Hannes’s and Anna’s children, which may have been present throughout their marriage. Still, these issues did not become apparent until Anna’s dementia began to worsen. Therefore, there were no family discussions about Anna’s care, and the role of the decision maker was unclear. Hannes and Anna’s children made Anna’s dementia care decisions in a vacuum. Had there been regular check-ins or meetings among the family members, previous health care decisions made and clearly documented, and ongoing communication amongst family members about Anna’s care, it is possible this case would still be difficult but not rise to the level of a wicked problem. But how were these family members to gain the skills and knowledge to navigate Anna’s changing dementia care needs? This question is also part of what makes their problem a wicked problem, because there were no healthcare, spiritual, or legal resources available that could help this family to understand, accept, and manage Anna’s care within the context of their family system.

Belonging to a blended family in later life when a spouse has dementia, calls for a multi-layered family system assessment for the caregiving spouse, the person with dementia, and their family/families. This case study shows that, at different stages of the dementia process, changes in family system dynamics can appear at the partnership, family, and system levels. Being in such chaos, as Hannes’s situation exemplified, can never be acceptable for anyone. Yet it is likely Hannes’s case is not singular.

### 4.1. The Partnership Level

As we examined Hannes’s and Anna’s partnership, we noted how different the outcome might have been if professionals had sat down with the family at the beginning of the diagnosis of Alzheimer’s disease to explain the process of dementia and potential care needs. As stated in Rittel and Webber’s [22] first explanation of wicked problems, Hannes’s and Anna’s family’s case has no clear definition of the problem, and perhaps none of the family members knew what was going to happen to Anna as the disease progressed. Professionals also did not understand what happened to make Hannes and his stepchildren’s relationship so complicated. However, this is a paradigmatic example of when a couple in a second marriage needs to cope with dementia and how family communication can fall apart. Hannes experienced many challenges before and after his wife moved to the nursing home. Our analysis has shown the chaotic events and disrespectful communication that arise in light of distrust and poor communication in the blended family system. It begs the question: who can handle such situations, especially for older adults with deteriorating health, declining cognitive ability, and declining cognitive literacy of place, time, and information? And how is the current healthcare system prepared to manage similar situations, given the increasing older population?

Many questions emerge from the story of Hannes’s and Anna’s marriage. Is a partnership in later years not considered as valuable in our health and social care system as when people are younger [4]. Is it only the first partnership that counts, or is it the partnership in which people have children that matters? This is a question worth considering when both spouses have children from previous partnerships and it becomes time to request institutional care for the spouse with dementia. When married, the caregiving spouse has the right to make legal decisions if the spouse with dementia is deemed incapacitated. However, in this case study, Anna’s children took away that right from Hannes. For professionals working on cases like this, we argue that it is essential to encourage the couple to discuss legal and financial considerations openly in the first stage of dementia. To be in a blended family formed in later life, it is also necessary for the couple to discuss and negotiate decisions about their care before the spouses become incapacitated [8]. Directives from different parties call for cooperation and support from the healthcare and social care systems, which may not currently be prepared or willing to get involved in family decision making.

### 4.2. The Family Level

This case study uncovered how all the tenants and ramifications of the family system emerged when professional care is needed, and family member’s relationships are under pressure. Professional staff and family members alike may need support to be able to navigate the challenges of family caregiving when dementia care is needed. We cannot overlook the partner and ignore the second partnership. This example raises the question of what could have been done for this family at the very beginning of the diagnosis and after the wife moved to a nursing home to prevent such destruction. We argue that early family therapy might have helped identify the potential emotional, legal and practical caregiving dilemmas or wicked problems. We can agree with Kettl [32] that our healthcare system is not equipped in all areas to respond to non-traditional and non-standard challenges such as this blended family. Families often need guidance to understand and define their rules for decision making. Often family caregiving rules are outside the healthcare professional’s awareness and therefore all family members may not be included or involved in care decisions [7,32]. Questions about who is responsible for making family decisions and caring for the person with dementia may arise. The answer Hannes got when he tried to consult with the healthcare staff was that they (as a family) had to solve their own problems.

Viewing this family through the lens of family systems theory, we can see how a family system can crumble when one member is diagnosed with dementia [31]. We wonder if early intervention and family assessment could have uncovered the emotional ties Anna’s and Hannes’s children had with each other and whether Anna’s children ever embraced Hannes’s and Anna’s marriage or pretended for the sake of their mother. We can learn from this case how important it is for health and social professionals to understand how family members of people with dementia define “family”. By asking simple questions like: Who is your family? What is your role in this family”? Are the family connections by blood, emotional or legal? Who is included or excluded from family decisions? Therefore, an open discussion of who has the decision making power is important early in the family dementia caregiving process.

### 4.3. The Healthcare System Level

This case study reveals that family caregiving for people with dementia can be influenced by family constellations and dynamics. Previous research has suggested that healthcare professionals currently have little education and few tools to support spouses and families of people with dementia [7]. This case study reveals that the current partner needs support and information to exercise his legal right to decide on matters of his wife’s care and the privacy of his home and property. Rittel and Webber’s definitions from seven to nine [22] in Table A1 show Hannes’s and Anna’s unique case that can be understood because of another problem, such as the impact of Anna’s previous partnership. Therefore, Hannes’s and Anna’s problems are intertwined and cannot be solved individually. Various professionals are needed to help and support them as a couple, maintaining the welfare of their blended family.

While this story is unique because of its complex and often tricky communications, it is not a single event. Healthcare providers from many disciplines will read this story and acknowledge similar stories from their practices and the difficulties in helping blended families through the transitions from ordinary partnership to one fraught with difficulties because of a spouse’s declining physical and mental health. Therefore, this is an intergenerational and interprofessional problem. Detailing Hannes’s family story has not yet given us any answers or a single solution. Still, it could describe ways we could work together to provide more support to spouses and family caregivers of older adults with dementia. And what can be done? Healthcare professionals must have an in-depth discussion about who is the next of kin and who should be included in health care decisions when a person suddenly or gradually develops dementia and becomes in need of care. When care decisions must be made in primary care settings, various primary care providers must be equipped with more information about what decisions spouses and families need to make at different stages of dementia to support the upholding of the whole family’s welfare.

As we have illustrated, the spouse will usually be considered the closest next of kin, today, however, there is a risk of being ignored or left in the shadows in the case of a second marriage. Sometimes children of the first marriage do not consider the new partner to be the next of kin; instead, they consider themselves in that position, even though the couple has been married for decades. It seems that today the healthcare system does not have clear ideas on how to meet cases like those illustrated in this article, such as couples in two or more marriages with blended families, and we risk violating the human rights of the individual family members, especially the spouses. Not only did Hannes lose his wife, his blended family, his privacy, and his property and wealth, but he also lost his dignity and his status as a partner by the professional who did not listen to him, and instead turned their attention to Anna’s children. The health care system is still not equipped to meet second-marriage couples and blended families and the potential complexities of dementia caregiving. Forgetting to delve into family histories in blended families is one reason there are conflicts and a lack of consensus about what the problem is, and which solution to choose. This calls for interdisciplinary perspectives and approaches in working with people with dementia, their spouses and their families [33].

## 5. Conclusions

In this case study, we have used “wicked problems” as an analytical lens, describing how a web of problems surrounding dementia care in blended families emerges. By mapping out these interconnected challenges, we have suggested a few promising practices to prepare and smoothen role transitions and family decision making in dementia care. Without trying to instigate or maintain collaboration and dialogue between professional and lay stakeholders, policymakers, and next of kin and extended families, we argue that blended families risk being transformed into wicked problems. We argue that the difference between single partnerships and multi-partnerships presents a difference that important in terms of care planning. The dearth of research on older age and multi-relationships and blended families is probably not due to lack of cases like the one we have analyzed, but rather, a need to acknowledge how lived life can re-emerge in unforeseen ways in older age. Next of kin, lay and professional care planners and carers can benefit from new knowledge about the intersections between previous and contemporary partnerships, next of kin, and extended and blended families.

## Figures and Tables

**Figure 1 ijerph-21-01213-f001:**
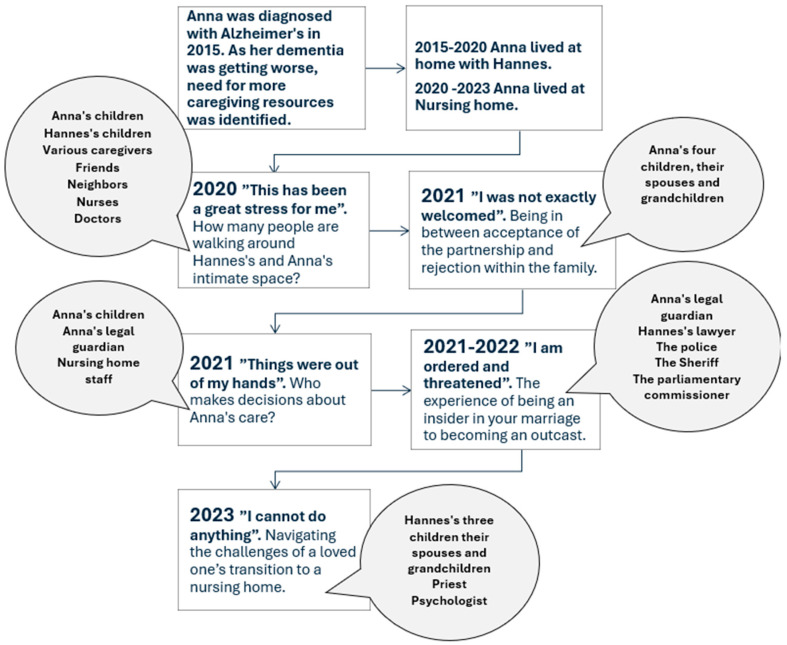
The timeline of Hannes’s and Anna’s family system dynamics from 2015–2023.

**Table 1 ijerph-21-01213-t001:** Rittel and Webber’s ten definitions of a wicked problem [22] (pp. 155–269).

There is no definitive formulation of a wicked problem. Wicked problems have no stopping rule. Solutions to wicked problems are not true or false but good or bad. There is no immediate and no ultimate test of a solution to a wicked problem. Every solution to a wicked problem is a “one-shot operation,” because there is no opportunity to learn by trial and error, every attempt counts significantly. Wicked problems do not have an enumerable (or an exhaustively describable) set of potential solutions, nor is there a well-described set of permissible operations that may be incorporated into the plan. Every wicked problem is essentially unique. Every wicked problem can be considered to be a symptom of another problem. The existence of a discrepancy representing a wicked problem can be explained in numerous ways. The choice of explanation determines the nature of the problem’s resolution. The planner has no right to be wrong.

## Data Availability

The data cannot be publicly available in repositories because they purport to sensitive issues. However, the data supporting the findings are available upon request from the corresponding author.

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
