# Peer review of "Second Partnership and Dementia Care in a Blended Family: Case Study of a Wicked Problem"

_ijerph, 2024, doi:10.3390/ijerph21091213_

Round 1

Reviewer 1 Report

Comments and Suggestions for Authors

The elaboration of the research question, made in the introduction, should be revised (as indicated in the note to the revised text, attached document)

Author Response

Thank you very much for taking the time to review our manuscript. The detailed responses to your comments are in the attached document, Table_respond_to_the_reviewers_comments.

You can find our response to you in Table I in that document.

Our revised submitted manuscript highlights all our revisions and changes in red. Hopefully, our answers in this table and the changes made in the manuscript parts are satisfying.

Reviewer 2 Report

Comments and Suggestions for Authors

I appreciate that you dive deeply into an individual case that is relevant for many similar family constellations, but get irritated by sloppy proof reading and some unnecessary shortcomings + that you would benefit from a careful reading by some qualified anglophone speaker. I also not happy with the word wicked - suggest evilness etc. How about complex? To mention some - I may have missed a number - shortcomings:.

47 Incorrect title of book (inconsistent capitals) and why give the title of it at all?

63 "it is called" >>  I assume stepchild is a pretty well-known concept?

68 families were just as complex in "the past", at least 100 years ago: She died, or he died, they remarried etc. See the Norwegian Folketeljinga 1801 for facts on this.

98 Why "" 

129 NH left unexplained

156 "those assets" - refers to what?

160 "con founders" ?

162 "intimate marriage life" sounds too intimate...? 

222-223 repetitious...

240-247 unorthodox interpunctions... ditto 298 & 440 i.a.

266 "know" >> recognize?

321-23 strange sentence    330 ditto

347 "and the Anna's children"     the?   Hanes misspelled---

414 "he them"  >> "he told them"?

427 "was this a --"  Don't understand

472    >> "challenging"?

478  some space before Table II please.

498-500 don't understand

513  "reveals THAT a challenge" ?

515 "leaves the question"  >>> "raises" ??

In the conclusions you suggest better communication... But the case just desribed may suggest that the animosities and hostilities were not negotiable or possible to solve by better communication?

Comments on the Quality of English Language

See above

Author Response

Thank you very much for taking the time to review our manuscript. The detailed responses to your comments are in the attached document, Table _ respond_to_the_reviewers_comments. 
You can find our response to you in Table II in that document. 
Our revised submitted manuscript highlights all our revisions and changes in red. Hopefully, our answers in this table and the changes made in the manuscript parts are satisfying. 
